**Subject Area:**
biochemistry/biotechnology/molecular biology/neuroscience

Alzheimer's disease, antioxidant, oxidative stress, neuroprotection, nutrient, nutraceutical

**Author for correspondence:**
Sandeep Kumar Singh
e-mail: sks.1247@gmail.com, sandeeps.bhu@gmail.com

# Role of antioxidants and a nutrient rich diet in Alzheimer's disease

Gerald Veurink[1,2,3], George Perry[4] and Sandeep Kumar Singh[3,5]

[1]Naturels, Armadale, Western Australia, Australia
[2]Department of Surgery, University of Western Australia, Perth, Australia
[3]Indian Scientific Education and Technology Foundation, Lucknow 226002, India
[4]Department of Biology, The University of Texas at San Antonio, San Antonio, TX, USA
[5]Centre of Biomedical Research, SGPGI Campus, Lucknow 226014, India

SKS, 0000-0002-0022-6240

The joint attack on the body by metabolic acidosis and oxidative stress suggests that treatment in degenerative diseases, including Alzheimer's disease (AD), may require a normalizing of extracellular and intracellular pH with simultaneous supplementation of an antioxidant combination cocktail at a sufficiently high dose. Evidence is also accumulating that combinations of antioxidants may be more effective, taking advantage of synergistic effects of appropriate antioxidants as well as a nutrient-rich diet to prevent and reverse AD. This review focuses on nutritional, nutraceutical and antioxidant treatments of AD, although they can also be used in other chronic degenerative and neurodegenerative diseases.

## 1. Introduction

Countless lives have been saved with antibiotics and vaccines for various communicable diseases [1,2]. However, chronic disease is currently the most significant burden on health systems globally and the cause of approximately 70% of deaths worldwide [3,4]. Alzheimer's disease (AD) and other neurodegenerative diseases have not specifically been included in those numbers and therefore the situation may be much worse. Approximately 45% of all Americans suffer from one or more chronic diseases [5].

In Europe, current estimates are that 50 million people live with multiple chronic conditions and this number is expected to increase during the next decade [6]. In 2015, dementia affected 47 million people worldwide (or roughly 5% of the world's elderly population), a figure that is predicted to increase to 75 million in 2030 and 132 million by 2050. Recent reviews estimate that globally nearly 9.9 million people develop dementia each year. People diagnosed with one or more chronic conditions often have complex health needs, die prematurely and have poorer overall quality of life [3]. Patients with multiple chronic conditions generally receive ineffective, incomplete and fragmented care [6].

According to the Grattan Institute, 'Australian primary care is failing in one crucial area: the prevention and management of chronic disease' [4], and it is probably the same globally. Professor Allen D. Roses has provided two excellent reviews on the economics and future of using pharmacogenetics to produce drugs with greater efficacy and safety on a more personalized treatment basis instead of limited efficacy for 30–40% of medicated patients [7,8]. However, it may take another 10 years or more for clinically trailed drugs to be brought to market. Medicinal drugs used to treat chronic illnesses need to be taken daily for the rest of each patient's life [9] and are therefore very profitable for the companies that produce them. These medicines, which provide symptomatic relief, are used in chronic diseases without the prospect of providing a cure. Unless there is a paradigm shift away from single-mode to multimodal medicines, or to combinations or cocktails of medicines which address all the factors of the disease process, it is unlikely that a cure will be found for chronic degenerative or neurodegenerative diseases

[10,11]. Focusing research efforts, drug development strategies and healthcare approaches predicated on a single component of a system, rather than the interacting network of components comprising such a system, may obscure important aetiological principles and/or disease mechanisms, including those evident during presymptomatic stages of disease. The application of systems science and its extension into healthcare therefore posits that health and/or disease result from the dynamic interactions of an individual's intrinsic multiomic components (e.g. genetic, epigenetic, etc.), their resultant phenotype, and the extrinsic (environmental) factors influencing the intrinsic milieu [12,13].

A holistic approach to healthcare to delay and prevent chronic disease by lifestyle changes that optimize individual diet, exercise, sleep and stress reduction is beneficial. Nutrition is also a central tenet of functional or integrative medicine, traditional Chinese medicine, Ayurveda and naturopathic medicine [12,13]. It is imperative that effective treatments for chronic diseases are implemented, reducing hospitalizations and serious complications to improve the quality of life of patients and lower the ever-increasing cost of healthcare, which we all share [5,6,14]. Research has demonstrated that a natural approach to preventing, delaying, and even reversing chronic degenerative and neurodegenerative diseases is effective [15–40].

In this review, we will focus on nutritional, nutraceutical and antioxidant treatments of AD, although they can also be used in other chronic degenerative and neurodegenerative diseases.

## 2. Alzheimer's disease

AD is the most common form of dementia in the aged and is characterized by cognitive decline and mental deterioration [41–43]. After heart disease and cancer, it is the third leading cause of death in the ageing population. The prevalence of AD is increasing exponentially with progressing age, affecting one in five people by the age of 80 [44]. AD is characterized histologically by the existence of intracellular and extracellular amyloid deposits in the brain. Beta amyloid (Aβ) is the major protein component of these deposits [45]. Aβ is a 4 kDa peptide which consists of 39–43 amino acids [46]. Aβ 1-40 is the major Aβ species and is soluble, whereas Aβ 1-42 is a minor soluble species and is fibrillogenic as exhibited in amyloid plaques. The Aβ peptide directly produces hydrogen peroxide through transition metal ion reduction, [47] thereby rendering it neurotoxic [48]. Increasing evidence has implicated Aβ in the induction of oxidative processes, either directly or indirectly, and this may have a key role in the neurotoxicity of the peptide. The neurotoxic mechanism of Aβ is being investigated although substantial evidence now exists which suggests that it exerts its effects through the production of oxygen free radicals. Butterfield [49] has demonstrated that Aβ interacts with the membrane lipids causing lipid peroxidation. As a consequence of lipid peroxidation the production of isoprostanes is increased. Since this adduct of lipid peroxidation is very stable it has been employed as a marker of lipid peroxidation [50]. Levels of isoprostanes were found to be elevated in the brains of AD patients. It has been shown that there is a 100% overlap between Aβ deposits and markers of oxidative stress in transgenic mice. This suggests that the transgenic mice with Aβ deposition show the same oxidative stress and damage response characteristic of AD. The association between oxidative stress and (Aβ) deposition possibly results in a positive

feedback system [51]. Tau aggregates are another characteristic of AD. The usually soluble tau becomes abnormally phosphorylated forming oligomers and larger filamentous aggregates. Hyperphosphorylated tau results in a clumping of filamentous actin forming neurofibrillary tangles, dysfunctional mitochondria and oxidative stress, and damages DNA, which thereby may cause cell death via apoptosis [52].

Since the aforementioned studies show that oxidative processes are important factors in neurodegeneration, it appears rational that antioxidants will be beneficial in the treatment of AD. Indeed, in a double-blind clinical trial, vitamin E at 2000 international units per day showed some beneficial effects with respect to the rate of deterioration of cognitive function [53]. Furthermore, in another double-blind placebo controlled clinical trial employing *Ginkgo biloba* extract (Egb 761), there was a slight improvement in cognitive function in Alzheimer's patients [54]. Further studies with more potent antioxidant combinations may prove to be more effective in treating this devastating disease due to their synergistic effect.

## 3. Oxidative stress

Oxidative stress is an imbalance between reactive oxygen species generation and antioxidants [55]. Oxidative stress can be caused by a variety of reasons, for example by an inadequate intake of antioxidants in the diet, or by the action of toxins in the body, such as smoking and pollution, or by inappropriate activation of phagocytes as in chronic inflammation. Oxidative stress has been implicated in numerous diseases. Tissue damage, by whatever insult or trauma, results in increased levels of free radicals and if antioxidant levels are minimal, then oxidative stress is the consequence. It is therefore important to assess the significance of oxidative stress in the disease process. The assessment will involve determining whether oxidative stress contributes directly to the disease process or whether it is merely an inactive end product.

Oxidative stress was initially proposed to be a major factor in AD in 1986 [56]. Since then many other researchers have found that oxidative stress is implicated in AD in various stages of the disease. Overwhelming evidence exists that the cells in the Alzheimer's brain undergo abnormally high levels of oxidative stress and that amyloid plaques are a focus of cellular and molecular oxidation. Oxidative stress is perhaps not the primary aetiology of AD; however, it precedes specific cellular and tissue damage, which underlies the onset of this disease [57]. Since 1994 various studies have established that oxidative stress is present in dying neurons and not just in (Aβ) deposits. Since oxidative stress is thought to play an important role in AD it follows that antioxidants may provide a useful therapy in the disease. There are many reasons and sources that cause oxidative stress in neurodegenerative diseases. Important sources of oxidative stress resulting in neurodegenerative disease are mentioned in figure 1.

## 4. Oxidative stress and decreased pH

Various studies have demonstrated that free radical formation is accelerated substantially as pH decreases (figure 2) [58–62]. For example, lipid peroxidation catalysed by transition metals such as iron is enhanced with decreasing extracellular pH. In fact, Fe2 and Fe3 are more soluble as solutions are more acidic, which makes them more available for lipid

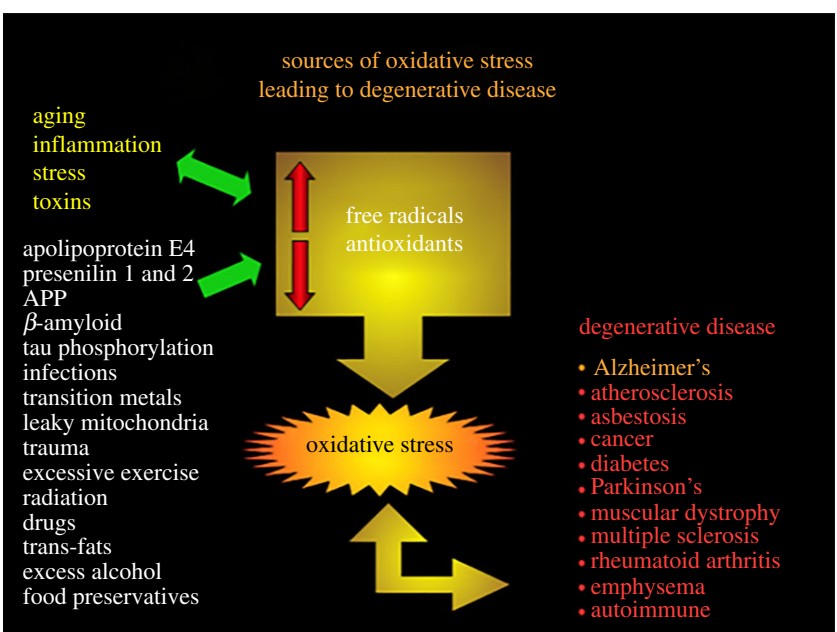

**Figure 1.** Sources of oxidative stress.

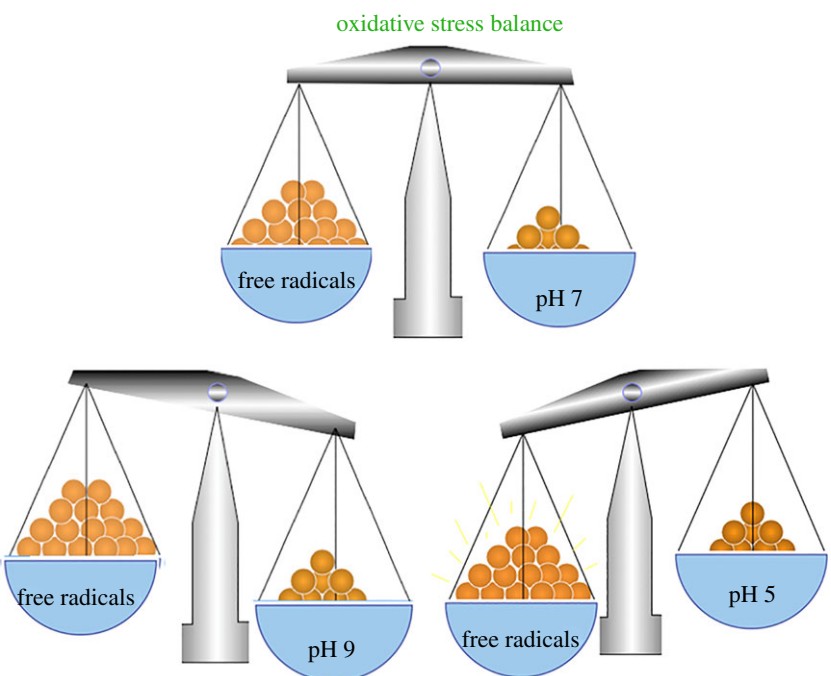

**Figure 2.** At pH 7 oxidative stress is relatively balanced, at pH 9 free radical production is reduced, at pH 5 free radical production is dramatically increased.

peroxidation reactions [62]. It has also been shown that an acidic pH can release iron from proteins such as lactoferrin [63,64] and ferritin [59]. In addition, cell culture studies have found that acidosis reduces the activity of antioxidant enzymes, which is in turn associated with increased intracellular free iron levels and increased oxidative stress [65]. Acidic pH releases iron from sequestered sites; this iron is more destructive due to enhanced solubility [62].

The problem is compounded by the fact that superoxide dismutation to hydrogen peroxide and oxygen occurs faster at an acidic pH [66] which could result in intensified free radical production. Ceruloplasmin, the main copper transport protein in plasma [67], contains six or seven copper atoms per molecule [68], one or more of which can partake in oxidation reactions depending on the degree of acidity [69].

Ceruloplasmin has been demonstrated to partake in antioxidant defence by inhibiting superoxide or ferritin-induced lipid peroxidation. Ceruloplasmin also scavenges hydrogen peroxide [70] and can act as a catalyst in the conversion of $Fe_2$ to $Fe_3$ [71]. A significant increase in the levels of ceruloplasmin has been found in the neuropil of AD brains [71], indicating that ceruloplasmin may indeed be instrumental in the production of free radicals in AD.

Decreased pH also results in the increased oxidation of lipids. For example, linoleic acid oxidation is increased by decreased pH [72], and the oxidation of polyunsaturated fatty acids occurred more rapidly at acidic pH [59,62,73]. In addition, homogenized liver and brain, both rich in lipids, oxidized more rapidly at acidic pH [59,73,74]. The above-mentioned studies provide strong evidence that decreased

royalsocietypublishing.org/journal/rsob    Open Biol. **10**: 200084

pH intensifies the production of detrimental free radicals. The situation is exacerbated by the commonly held view that pH is maintained at an appropriate level, since blood pH is well maintained. Chronic metabolic acidosis, which is associated with intracellular and extracellular acidosis, can exist even though blood pH is maintained within a normal range [75]. While it is generally understood by most physicians that severe disorders of acid–base metabolism are dangerous to the patient, the dangers of mild metabolic acidosis are less appreciated [76–78].

In a study on the relationship between glucose and brain lactate levels during cerebral ischaemia in gerbils, it was shown that intracellular pH was markedly decreased when brain lactate levels rose above 17 $\mu$mol g$^{-1}$ [79]. Ageing and excessive dietary protein and meat intake are associated with chronic metabolic acidosis; however, these factors are often overlooked since blood pH appears normal [75,80,81]. The association of metabolic acidosis with ageing may also be complicated by a decreased rate of renal acid excretion associated with a progressive loss of nephrons, resulting in a reduced glomerular filtration rate [82]. A study employing the 31P-MRS-based measurement of intracellular and interstitial pH *in vivo* has shown that neurons have a basal pH of 6.95 and astrocytes a pH of 7.05, and that they are capable of substantial regulation of intracellular pH, despite interstitial pH decreasing by 0.31 pH units [83]. Nevertheless, evidence of the detrimental effects of decreased pH has been substantiated in *in vitro* studies. Neurons isolated from the hippocampus of aged rats are more susceptible to lactic acid induced toxicity [78], and brain capillary endothelial cells as well as cholinergic neurons were shown to be vulnerable to low pH [84]. It has also been demonstrated that astrocytes subjected to an acidic environment show rapidly increased glial fibrillary acidic protein (GFAP) immunoreactivity [85]. In another study, lymphoblasts from AD patients exhibited a lower H$^+$-buffering capacity and a decreased rate of H$^+$ removal when subjected to an acid load in comparison to control cells [86]. Moreover, in an *in vivo* study it was demonstrated that acute acidosis elevated malonaldehyde in rats [87]. Acidosis-induced swelling and intracellular acidification of glial cells have been demonstrated [88], suggesting that this may be the mechanism by which glia and astrocytes become reactive.

Further evidence for a role of acidosis in AD comes from studies which have demonstrated that Aβ aggregates more avidly at pH 6.8 than pH 7.4 [89], and incubation of Aβ with Fe$_3$-Citrate or Cu2-Glycine at pH 6.8 resulted in the production of more Fe2 and Cu1 than at pH 7.4 [47,90]. Adding to this is evidence of the formation of larger and more complex fibrils from Aβ at acidic pH which was demonstrated when undifferentiated rat pheochromocytoma (PC12) cells were subjected to a pH 5.8 environment compared to pH 7.4 [91]. Embryonic rat hippocampal neurons incubated in serum-free neurobasal medium were shown to lead to an increase in Aβ immunoreactivity when subjected to lactic acid [76]. *In vitro* studies have also shown that the β-secretase ASP-2 cleaves the APP at pH5 but not at pH 8.5 [92]. This may support the theory that Aβ peptides are produced in acidic organelles, since Aβ peptides have been found in low-pH organelles like the endosomes and lysosomes [93].

Lactate is increased and pH values are decreased in human postmortem brains from patients who have died in an agonal state [94,95]. The assessment of AD and Down syndrome brains has also shown a decreased pH and increased lactate levels compared to controls [96]. Brain levels of lactic acid have also been demonstrated to rise sharply during ischaemia as a result of decreased blood flow to the brain [97]. Early onset AD patients have been found to have a fourfold increase in lactate [98], suggesting that Aβ is accumulating at low-pH, although it is not known if the lower pH is a result of accumulating Aβ in early onset AD patients, or a precursor to Aβ accumulation. Apoptotic neurons in AD brains frequently display intracellular Aβ42 labelling [99]. There is evidence that neurons accumulating Aβ undergo lysis to form amyloid plaques [100]. Aβ plaques may represent the redox silencing and entombment of Aβ by the transition metal zinc [101]. Investigations with substances known to promote a more alkaline cellular environment, such as potassium citrate and calcium carbonate, are warranted. These may improve total antioxidant status through the mechanism of reduced production of free radicals. Indeed, recently it has been shown that long-term intake of a high-protein diet modulated acid–base metabolism, which was neutralized by dietary supplementation of potassium citrate in male rats [102]. Furthermore, in a randomized, prospective, controlled, double-blind trial, postmenopausal women with osteopenia were shown to have increased bone mass after potassium citrate treatment as well as decreased blood pressure [103]. Ageing is associated with increased free radical production; therefore it is conceivable that age mediates an increased sensitivity to low pH, triggering increases in oxidative stress.

In Veurink's unpublished research employing an ISFET pH mV$^{-1}$ meter it is demonstrated that there is a very close relationship between pH and mV or oxidation reduction potentials (figures 3 and 4).

Taken together, this suggests that antioxidant status of biological fluids and tissue homogenates may be improved just by increasing pH to a more alkaline level. Thus it is also conceivable that, by improving extracellular and intracellular pH, the antioxidants supplied by an optimal diet are perhaps sufficient to complement the antioxidant system of the body in order to prevent disease. However, further research employing potassium citrate or potassium bicarbonate would be needed to test that hypothesis.

## 5. Effect of diet on extracellular pH

There has been a profound transformation of the human diet consequent to agricultural development, animal husbandry and the development of modern food production methods. The contemporary diet has an overabundance of unhealthy fats, sugar and sodium chloride, and a paucity of fibre, calcium and potassium [104–107]. It has been estimated that in the palaeolithic diet, sodium intake was at about 29 meq and potassium intake in excess of 280 meq per day. By contrast, modern humans consume between 100 and 300 meq of sodium and about 80 meq of potassium per day [108]. As a consequence of this dietary transformation, contemporary humans are not only overloaded with sodium and chloride but are also deficient in potassium and bicarbonates. Therefore, from a relatively young age through to old age, humans may develop a progressive increase in extracellular acidity and decrease in plasma bicarbonates. Together, these are indicative of increasing low-grade metabolic acidosis. However, increasing dietary potassium to levels estimated in the

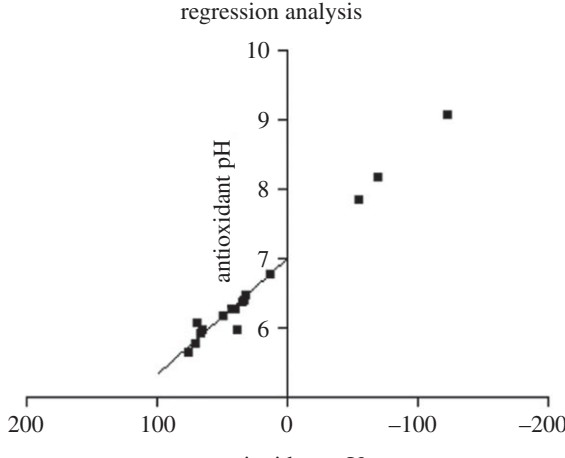

**Figure 3.** A graphical representation of antioxidant solutions mV versus antioxidant pH regression analysis. Statistical analysis using linear regression in Graphpad Prism 3 for Windows 95/98 demonstrated a significant relationship between antioxidant mV and antioxidant pH ($r^2 = 0.9840$).

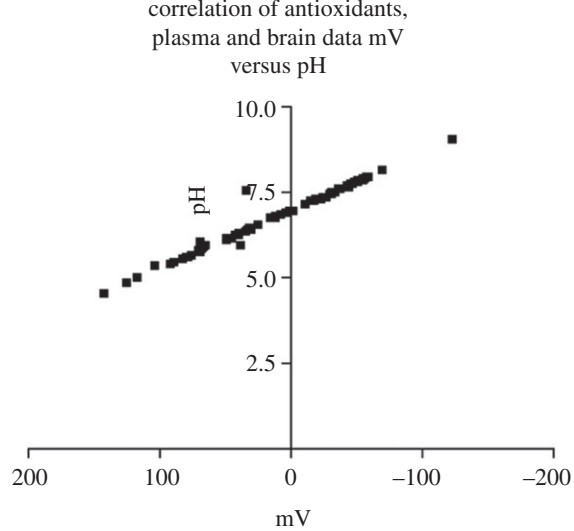

**Figure 4.** A graphical representation of the correlation of mV versus pH of all antioxidants, plasma and brain data. Statistical analysis using Graphpad Prism 3 for Windows 95/98 demonstrated a very strong relationship between mV and pH ($r^2 = 0.9822$). Furthermore, this was shown not only in antioxidant solutions but also in herbal extracts, fruit juices, wine, biological fluids and homogenised tissues. The results suggest that a simple pH mV$^{-1}$ assessment of antioxidant solutions, biological fluids and tissue homogenates can be used to give a reasonably accurate indication of the total antioxidant capacity or redox potential.

palaeolithic diet, by eating more fruits and non-grain plant foods, may hold benefits for preventing or delaying many of the diet- and age-related degenerative diseases and their consequences [109,110].

# 6. Metabolic acidosis

Diet may be one of the main contributing factors to low-grade metabolic acidosis leading to increased all-cause mortality and chronic diseases such as type 2 diabetes mellitus and hypertension, which are also associated with AD. Generally, foods which tend to decrease pH are from animal origin and foods which raise pH are mostly of plant origin [111]. Metabolic acidosis has been demonstrated to increase protein breakdown in humans. Moreover, it also stimulates branched chain amino acid oxidation in both humans and animals [112]. Thus, metabolic acidosis could result in increased oxidative stress as well as protein breakdown and aggregation, thereby contributing to a worsening of the disease state. It has also been established that, during ischaemia, intracellular pH of neurons and glia generally acidify to pH 6.5, and during trauma, brain pH acidifies to pH 6.2–6.8. If hyperglycaemia precedes ischaemia, then pH can become as low as pH 6.0. Experimental research undertaken has also shown that acidosis by different means ultimately results in neuron loss [113,114]. Dietary metabolic acidosis by modulating cortisol output may influence risk for insulin resistance syndrome (see review by McCarty [115]). It is also suggested that there is a strong link or association between insulin resistance and AD [116,117]. Furthermore, in their study showing acute acidosis elevates malonaldehyde in rat brain [87], Waterfall *et al.* have provided *in vivo* evidence for acidosis-induced oxidative stress in brain tissue. In addition, in an *in vitro* study, it was shown that acidic pH promotes the formation of toxic β amyloid fibrils and that they induced significant apoptotic death of rat PC12 cells [91]. Therefore, metabolic acidosis, together with insulin resistance and oxidative stress, may severely impact the development and progression of AD.

Considering the multifactorial nature of AD and the factors discussed above it may suggest that AD, like other neurodegenerative diseases and likely all degenerative diseases, may have a common link. Namely that by dietary intake, age-dependent metabolic acidosis is incurred which causes oxidative stress, ultimately leading to the development and cyclical continuation, through a feedback loop, of disease in various regions of the human body. Of course, the degenerative disease process is also impacted upon by environmental factors, the ageing process and genetic disposition. The joint attack on the body by metabolic acidosis and oxidative stress suggests that treatment in degenerative diseases, including AD, may require a normalizing of extracellular and intracellular pH with a simultaneous supplementation of an antioxidant combination cocktail at a sufficiently high dose. Various studies have shown that dietary intake of fresh fruit and vegetables is very effective in reducing or halting oxidative stress. The reason that many studies wherein supplementation of antioxidants is used to remove oxidative stress are not that convincing is that perhaps the normalizing of the acid-base was not considered nor dealt with. Moreover, the dietary intake of fresh fruits and vegetables help to normalize pH due to the high levels of alkaline minerals contained in them. Alternatively, the intake of antioxidants may have been 'a little too low and a little too late'. Furthermore, it may be that dietary antioxidant intake from a well-balanced diet consisting of a high percentage of fresh fruit and vegetables may be sufficient to prevent or deal with disease when extracellular and intracellular acid-base is normalized and maintained by supplementation with potassium citrate, fresh fruits and vegetables.

In Veurink's unpublished data 12-month-old Tg2576 transgenic mice were fed diets with or without antioxidant supplementation in an attempt to assess the effect of large doses of antioxidants on the development of AD-like neuropathology and memory deficits. The antioxidants included vitamin E acetate, vitamin C palmitate, *Ginkgo biloba* and grape seed extract

royalsocietypublishing.org/journal/rsob    *Open Biol.* **10**: 200084

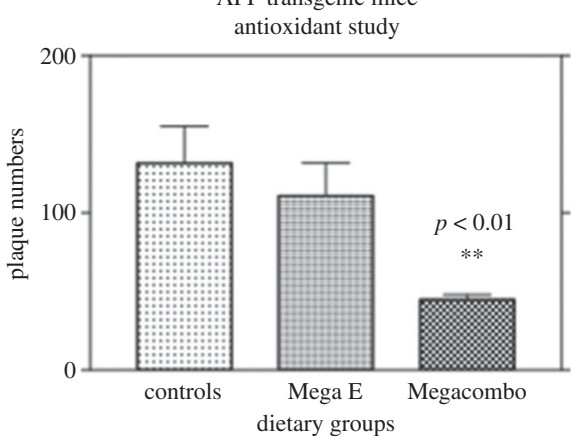

**Figure 5.** The graphical representation of the effects of standard chow and antioxidant supplementation for three months on mean plaque numbers in the brains of 15-month-old female APP transgenic mice. Controls were APP transgenic mice fed Std Chow ($n = 7$), Mega E were APP transgenic mice fed Std Chow supplemented with a high dose of vitamin E acetate at 64.8 gm kg of feed ($n = 6$), Megacombo were APP transgenic mice fed a combination of antioxidants including vitamin E acetate at 28.8 gm kg$^{-1}$ of feed vitamin C palmitate at 28.8 gm kg$^{-1}$ of feed, *Ginkgo biloba* at 3.6 m kg$^{-1}$ of feed and grape seed extract (Pycnogenol) at 3.6 gm kg$^{-1}$ of feed ($n = 6$). Statistical analysis using a Kruskal–Wallis non-parametric ANOVA demonstrated a statistically significant difference between the control group and the megacombo group ($p < 0.01$).

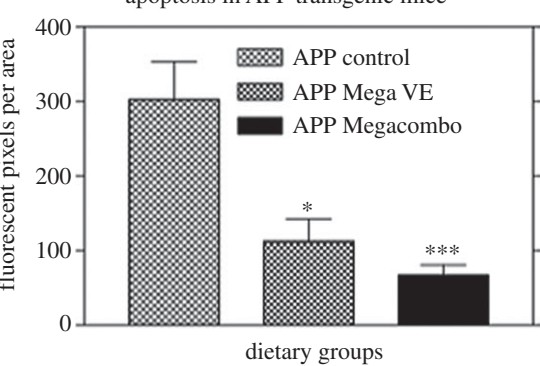

**Figure 6.** A graphical representation of the effect of antioxidant supplementation on apoptosis in APP Transgenic mice. Controls were APP transgenic mice fed Std Chow ($n = 8$), Mega E were APP transgenic mice fed Std Chow supplemented with a high dose of vitamin E acetate at 64.8 kg of feed ($n = 6$), Megacombo were APP transgenic mice fed a combination of antioxidants including vitamin E acetate at 28.8 gm kg of feed, vitamin C palmitate at 28.8 gm kg of feed, *Ginkgo biloba* at 36 gm kg$^{-1}$ of feed and grape seed extract Pycnogenol) at 3.6 gm kg$^{-1}$ of feed ($n = 8$). Statistical analysis using a non-parametric ANOVA with a Dunn's multiple comparisons test in Graphpad Prism 3 for Windows 95/98 showed a statistically significant difference between the APP control group versus the APP Mega VE group ($p < 0.05$) and also between the APP control group versus the APP Megacombo group ($p < 0.001$).

(pycnogenol) supplemented to a Standard (Std) Chow diet. Tg2576 mice use the prion protein promoter to express AβPP with the Swedish double mutation (K670N/M671 L, AβPPSwe), resulting in increased total Aβ production [118]. These mice deposit Aβ plaques and some vascular amyloid, and develop neuritic dystrophy and gliosis at 6–10 months. They also demonstrate progressive behavioural and cognitive deficits.

Aβ plaque deposition, levels of apoptosis and memory performance were investigated in these mice following three months on the different diets. Tg2576 transgenic mice were fed large doses of either vitamin E or large doses of an antioxidant combination, and were compared to mice fed a Std Chow diet. Twenty-four female 12-month-old Tg2576 mice were given either a Std Chow diet without additives, a Std Chow diet supplemented with a high dose of vitamin E acetate at 64.8 g kg$^{-1}$ of feed, or a Std Chow diet supplemented with a combination of antioxidants that consisted of vitamin E acetate at 28.8 g kg$^{-1}$ of feed, vitamin C palmitate at 28.8 g kg$^{-1}$ of feed, *Ginkgo biloba* at 3.6 g kg$^{-1}$ of feed and grape seed extract (pycnogenol) at 3.6 g kg$^{-1}$ of feed. The mice were maintained on these diets for a period of three months.

The results (figure 5) demonstrate that supplementation with large doses of antioxidants was instrumental in reducing the number of plaques in the brains of Tg2576 AD transgenic mice. Although vitamin E supplementation showed a trend toward decreasing numbers of plaques, this was not significant, unlike the result obtained in the mice fed the combination of antioxidants. The results from the work described above indicate that antioxidant supplementation at high doses can be effective in slowing down, stopping, or even reversing Aβ plaque deposition in Tg2576 transgenic mice.

Secondly, antioxidant supplementation at high doses was demonstrated to be effective in reducing apoptosis associated

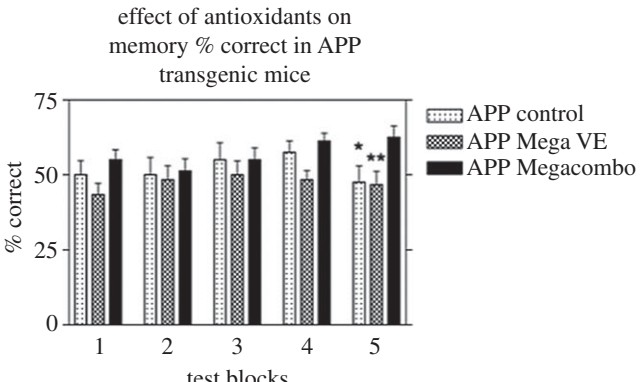

**Figure 7.** A graphical representation of the effect of antioxidants on memory % correct in APP transgenic mice. Statistical analysis using an ANOVA in Graphpad Prism 3 showed a statistically significant difference between the APP Mega VE and the Megacombo groups ($p < 0.01$). Using an unpaired *t*-test showed a statistically significant difference between the APP control and the APP Megacombo in block 5 ($p < 0.05$).

fluorescent pixel areas (figure 6), again supporting the hypotheses that oxidative stress is of primary importance in the neuropathological pathway of Aβ-induced damage in the AD brain, and that antioxidant supplementation can alleviate some of the damage produced by excessive Aβ production. These findings are consistent with other studies which have demonstrated that a combination of antioxidants can be effective in reducing apoptosis [38].

The results (figure 7) indicate a significant improvement in memory in the transgenic animals fed Std Chow supplemented with high doses of the antioxidant combination when compared to the other transgenic mice fed Std Chow alone or Std

**Table 1.** Antioxidants and dietary interventions in Alzheimer's disease.

| antioxidants and nutraceuticals | effect or biological action | references |
| --- | --- | --- |
| molecular hydrogen | An exceptional antioxidant which also reduces inflammation and modulation signalling pathways. Diffuses into mitochondria and nucleus reacting with free radicals act their source. Reduces the hydroxyl radical. Reduces beta amyloid-induced ROS. accumulation. Suppressed learning and memory impairment and extended lifespan. Improved word recall scores. Increases superoxide dismutase and glutathione levels. | [119–121,124–126] |
| glutathione | Maintains the thiol redox status of cells, protects against oxidative stress, detoxes reactive metals and electrophiles. Scavenges lipid peroxidation products. Beneficial in maintaining good health in the aged. | [126–129] |
| astaxanthin | Most potent carotenoid having neuroprotective properties. Reduces oxidative stress, inflammation and apoptosis. Protects against the neurotoxic effects of beta amyloid oligomers. Decreased memory impairment in Wistar rats. | [130–134] |
| ascorbyl palmitate | Maintains vitamin C activity without the side effects of ascorbic acid. Is an efficient scavenger of the hydroxyl radical. Able to cross the blood brain barrier. Regenerates vitamin E. | [135–143] |
| nicotinic acid (niacin) (vitamin B3) | Megadoses of nicotinic acid restored mental capacity in previous prisoners of war. | [144] |
| vitamin B12 and B9 (folate) | Reduces homocysteine | [145] |
| fruits and vegetables | Dietary supplementation with fruits, vegetables and their extracts can decrease oxidative stress and inflammatiom | [146] |
| dietary restriction, Mediterranean diet, lifestyle changes | The evidence supports nutritional interventions and lifestyle changes prevent and treat Alzheimer's disease. | [15,147,148] |
| low-carbohydrate diet | Reduces triglycerides, blood glucose and insulin resistance common in Alzheimer's patients. | [149–152] |
| ketogenic diet and lifestyle changes | Reverse memory problems in 100 Alzheimer's patients. | [19] |

Chow with vitamin E as the sole extra antioxidant. The results also suggest that high doses of the combination of antioxidants helped the transgenic mice learn where the water baits were placed in the maze just as well as the control, non-transgenic mice fed Std Chow alone. In these studies, and from our small study of the individual antioxidant vitamin E as well as vitamin E combined with other antioxidants, evidence is accumulating that combinations of antioxidants may be more effective, taking advantage of synergistic effects of appropriate antioxidant combinations.

# 7. Antioxidants

As already mentioned oxidative processes are important factors in neurodegeneration, so it appears rational that antioxidants will be beneficial in the treatment of AD.

An antioxidant is a compound which reacts with free radicals to render them harmless. The term is often used to describe chain-breaking inhibitors of lipid peroxidation. Yet, free radicals in addition to lipid peroxidation also damage proteins, DNA, and almost any type of biomolecule. Extensive research has shown that oxidative stress is an early factor in the AD process and that antioxidants have minimized their deleterious effects. For a very comprehensive review see [35,119,120]. In considering these reviews, we may conclude that antioxidants have an important function in neuroprotection and therefore we will examine some of the most effective antioxidants.

# 8. Molecular hydrogen antioxidant

Molecular hydrogen ($H_2$) has been researched recently for various oxidative stress-related diseases [121–123] (table 1). It is an exceptional antioxidant which reduces inflammation and modulation of signalling pathways, thereby providing cytoprotection [153]. By virtue of the fact that molecular hydrogen is the lightest gas and the smallest molecule in existence, it readily penetrates cell membranes and lipid bilayers, and diffuses into the cellular organelles such as the mitochondria or the nucleus, thereby reacting with free radicals at one of their major sources. Molecular hydrogen can be administered by various methods, including inhalation, ingesting hydrogen-rich water, injecting hydrogen rich saline, bathing in hydrogen-rich water or by increasing production of intestinal hydrogen by bacterial effect on undigestible carbohydrates. Moreover, the reactivity of molecular hydrogen is so mild that it does not react with physiological relevant reactive oxygen species which are involved in cell signalling or defensive mechanisms against microbes. See reviews in [121–123,153]. Molecular hydrogen has been shown to selectively reduce the most destructive hydroxyl radical implicated in the destruction of nucleic acids, proteins and causing lipid peroxidation which is also characteristic in AD [124]. Administration of molecular hydrogen to short-lived *Drosophila* increased their survival and life span [125]. Hydrogen rich water attenuated (Aβ) induced neurotoxicity in cultured human neuronal cells, upregulated (Aβ) suppressed AMPK and downstream Sirt1-FoxO3a signalling, reduced

**Figure 8.** Figure showing the chemical structure of (*a*) glutathione, (*b*) astaxanthin and (*c*) ascorbyl palmitate.

(Aβ)-induced reactive oxygen species accumulation, and upregulated intracellular anti-oxidative enzymes such as superoxide dismutase 1 & 2 and catalase [127]. In a mouse model hydrogen rich water was shown to suppress a decline in learning and memory impairment, extending the average lifespan [154]. Furthermore, in a subsequent randomized placebo controlled clinical study in APOE4 genotype patients, the total ADAS-cogs and word recall task scores were significantly improved after one year of drinking hydrogen-rich water [154]. These research studies reveal that molecular hydrogen may have great potential for suppressing AD.

# 9. Glutathione antioxidant

Glutathione is the most important endogenous antioxidant, with the chemical structure shown in figure 8*a*. It is needed for maintaining the thiol redox status of cells, protection against oxidative stress, detoxing reactive metals and electrophiles. Glutathione is also needed for the storage and transport of cysteine, and for protein and DNA synthesis, cell cycle regulation and cell differentiation [155]. Glutathione is also an excellent scavenger of lipid peroxidation products including 4-hydroxy-2-nonenal (HNE) and acrolein which bind proteins, thus inhibiting their normal activity. [155]. Glutathione is also extremely important in that it forms metal complexes via non enzymatic reactions with metal ions such as arsenic, cadmium, copper, gold, lead, silver, mercury and zinc so that they can be eliminated from the body in a detoxification process [155–157].

A higher level of glutathione in centenarians was found to be associated with the best functional capacity, suggesting that an increased level of glutathione is beneficial in maintaining good health [128]. Glutathione levels may be increased through eating specific foods and nutrients or antioxidant supplements to maintain optimal amounts. Humans require the amino acids glycine, cysteine and glutamic acid in order to produce sufficient levels of glutathione. It has been suggested that cysteine, which is a sulfur amino acid, may be ingested in sulfur-rich foods to increase glutathione synthesis. *N*-acetylcysteine (NAC), itself an antioxidant and rich in cysteine, is suggested as a supplement to increase glutathione. Whey protein concentrate, Omega-3 fatty acids, salmon, vitamin B complex, vitamin C, vitamin E, alpha-lipoic acid, selenium, phytonutrients, citrus fruits and cruciferous vegetables rich in sulforaphane may be used to increase glutathione levels [129]. Apple cider vinegar has also been shown to increase the activity of the antioxidant enzymes, superoxide dismutase, catalase and glutathione peroxidase, and it reduced lipid peroxidation [126,130]. Molecular hydrogen in the form of hydrogen rich water was shown to increase superoxide dismutase as well as glutathione levels in young healthy males [131].

# 10. Astaxanthin

Astaxanthin is the most potent carotenoid antioxidant. It is lipid soluble belonging to xanthophylls which have been demonstrated to have neuroprotective properties. Astaxanthin may be sourced from shrimp, asteroidean, algae, lobster, crustacean, krill, trout, red sea bream and salmon. It is mostly isolated from the microalgae *Haematococcus pluvialis* [132]. Astaxanthin has a linear polar-non-polar-polar structure with keto and hydroxyl moieties at the polar ends, and has conjugated carbon–carbon double bonds at a non-polar middle part (figure 8*b*) that enables it to fit specifically into the same span of cell membranes as well as be able to pass through the blood-brain barrier. Research studies have demonstrated that astaxanthin is effective in reducing oxidative stress, inflammation and apoptosis, which are key factors in the process of neurodegeneration. Researchers have shown that Astaxanthin reduced ischaemia-associated injury in brain tissue by inhibiting oxidative stress and protected neuroblastoma cells against Aβ-induced oxidative stress. Astaxanthin was shown to protect primary hippocampal neurons from the neurotoxic effects of Aβ oligomers, thereby supporting the notion that daily consumption of Astaxanthin may be beneficial in AD as well as other neurodegenerative diseases [133].

In a study of the role of Astaxanthin in hippocampal insulin resistance induced by Aβ peptides in Wistar rats, the results demonstrated a dose dependent reversal of memory impairment. [134]. These and other studies indicate that Astaxanthin is a promising antioxidant for the treatment of neurodegenerative disorders including Alzheimer's and Parkinson's disease [15,158]. Astaxanthin mitigates oxidative stress in various neurodegenerative disorders by preventing oxidative stress induced mitochondrial dysfunction (see review [135]). Further research is under way in order to improve the bioavailability of Astaxanthin [136].

royalsocietypublishing.org/journal/rsob    Open Biol. **10**: 200084

## 11. Ascorbyl palmitate

Oral supplementation of vitamin C may be particularly desirable in humans, since humans are not able to synthesize vitamin C like many other animals [137]. Ascorbyl palmitate (also known as L-ascorbyl-6-palmitate; 6-O-palmitoylascorbic acid or L-ascorbic acid, 6-hexadecanoate; figure 8c) is a fat-soluble synthetic derivative or analogue of vitamin C (ascorbic acid).

Ascorbyl palmitate maintains all the antioxidant activity of vitamin C without the problems that can be associated with ascorbic acid, the water-soluble form of vitamin C [138]. Tissue demand for vitamin C is better satisfied when it is supplied in its lipophilic rather than in its hydrophilic form [136]. Extensive use has been made of this antioxidant in foods, pharmaceuticals and skin care products to prevent the oxidation of various oils and waxes, and research has established that it is an efficient hydroxyl free radical scavenger [139–143,159]. Being amphipathic allows it to concentrate into the phospholipid membranes in biological systems whereby the fatty acid portion is intercalated into the outer layer of the bi-layered membrane and the inner portion (ascorbate head) is buried into the inner membrane [138,160]. It is also stable at neutral pH. Ascorbyl palmitate can therefore be active inside as well as outside the cells [137].

The FDA status is GRAS (generally recognized as safe) and there is no limitation on levels which can be used in food or cosmetics; it, therefore, could prove to be an ideal agent for the protection of cell membranes which generally are very susceptible to free radical attack via lipid peroxidation [143]. Ascorbyl palmitate is able to cross the BBB and thus, on average, makes more vitamin C available to neural tissue by an order of magnitude [137]. Furthermore, because it resides in the cell membrane, ascorbyl palmitate can regenerate the vitamin E radical continuously, unlike ascorbic acid which only regenerates the vitamin E radical at the interface of water-soluble and lipid components. Many cross-sectional large-scale and long-term studies have tried to establish whether the use of vitamin C supplements, either alone or in combination with other supplements such as vitamin E, reduce the incidence of AD. Some studies suggest vitamin C supplementation does reduce the incidence of AD [161]; however, some studies produced ambiguous results, and further long-term studies are required [162]. In a randomized study of various antioxidants on their effect on inclusion bodies in the brains of apoE-deficient mice, the group receiving ascorbyl palmitate had the least number of inclusion bodies (G.V. 2007, G.P. 2020, S.K.S. 2020, unpublished data). That suggests that ascorbyl palmitate crosses the blood–brain barrier and may be an antioxidant to consider in future research in AD.

However, some studies on single use antioxidants have not demonstrated efficacy. It may be that different forms of vitamin E including the four tocopherols as well as the toco-trienols when combined may be more effective since high doses of alpha tocopherol can decrease the bioavailability of the other forms potentially having a detrimental effect. Moreover high doses of vitamin C have been posited to have a pro-oxidant effect in some studies. For this reason, it may be advantageous to assess the oxidation reduction potential of patients and dose antioxidants at personalized levels.

Future clinical trials should incorporate cocktails of antioxidants which have a synergistic and antioxidant recycling activity [163].

## 12. The Keap1-Nrf2-ARE pathway

The Keap1-Nrf2-ARE pathway recently has been receiving attention since it plays an important function in protecting cells from oxidative stress by activating Nrf2, to induce the downstream phase II enzymes such as heme-oxygenase-1, superoxide dismutase, glutathione peroxidase, glutamate-cysteine ligase, catalase and others. These enzymes are not consumed by their antioxidant actions and they catalyse many chemical detoxification reactions and some regenerate small molecule antioxidants. This may be an important target for the potential development of novel therapeutic agents to treat various degenerative and neurodegenerative diseases [144,164].

## 13. Dietary interventions for Alzheimer's

The concept that diet can affect mental ability and susceptibility to neurological disorders is not new. Several thousand soldiers held as prisoners of war in Japanese camps were made prematurely senile by almost four years of malnutrition. Supplementation with megadoses of nicotinic acid (3 grams per day) restored mental capacity. This led to the conclusion that 'senility' is due to chronic malnutrition and that it is a vitamin-dependent condition which comes from many years of mild or moderate chronic vitamin deficiencies [145]. A variety of other nutritional factors have also been integrally linked with AD (see reviews in [146–148]). For example, it has been shown that dysregulation of energy balance, vitamin B12, folate and homocysteine levels plays a role in the pathogenesis of AD.

It has been suggested that an integrated medicine approach combining evidence-based treatments from the literature on dietary intervention, a reduction in stress, an increase in exercise, and dietary supplementation with pharmaceuticals and/or vitamins and antioxidants into an all-embracing complementary treatment strategy would benefit the elderly in many health aspects, and possibly reduce the risk of age-related conditions including AD [28,147,148].

## 14. Mineral and antioxidant deficiencies in foods

### 14.1. Agricultural methods

The nutritional value of crop plants is determined by a number of factors including genetic makeup, the type of soil in which the plants are grown, seasonal effects, stage of maturity at harvest, and the quantity and type of fertilizers used in the production of the plants [165]. Hundreds of years of agriculture using the same surface soil in many countries have slowly drained soils of minerals. In addition, pesticides and herbicides have been sprayed on most soils, inadvertently destroying the microorganisms which are needed to release many of these minerals and to maintain soil fertility [166].

In Western Australia as elsewhere, it is common practice to test soils for nitrogen, phosphorus and potassium (NPK) levels; other macro and trace elements are rarely tested, and thus depletion of many essential minerals has not been documented carefully, or even measured, in many areas. The continuous yearly application of NPK fertilizer has resulted in a high phosphate status of soils, and two-thirds of high phosphate status soils have been found to be deficient in sulfur and a quarter

deficient in potassium [167]. Problems are compounded by the fact that high phosphate soils greatly diminish the uptake of copper, zinc and manganese by plants, probably owing to the formation of complex phosphate salts [168].

## 14.2. Trace or essential minerals

Essential minerals and trace elements are not replenished with the addition of NPK fertilizer and the resulting food crops are severely deficient in them. Low zinc levels in soils have also been found in many regions throughout the world: this may directly impact on antioxidant status since zinc has a direct antioxidant action by occupying iron or copper binding sites in lipids, proteins and DNA [169–171]. Interestingly, it has also recently been reported that zinc binding to Aβ inhibits neurotoxicity by suppressing the generation of hydrogen peroxide [101]. These studies and many others suggest there is a need for mineral replenishment of soils, since depleted soils cause deficiencies in plants which may have a detrimental impact on the entire food chain. For example, studies comparing the antioxidant status of fruits such as plums, pears and peaches have found that levels of vitamins C and E and polyphenols are significantly higher in these fruits when grown using organic practices instead of conventional methods [172,173]. It has also been demonstrated that increased soil organic matter content increases the uptake of copper, zinc and manganese in oat crops [168]. The agricultural problems are further aggravated by the common harvesting procedure of picking fruits and vegetables before they reach maturity, despite the fact that most fruits and vegetables reach their maximum vitamin content at maturity. This practice also impacts their phytonutrient and antioxidant content [174].

## 15. Food processing and storage methods

Extensive research needs to be undertaken on the effects of storage on vitamin and phytonutrient levels in fruits and vegetables, since some studies have demonstrated their reduction during storage [175]. Processing and preservation of foods increases the problem further; for example, nutrient losses following the refining of flour and sugar have been demonstrated. Similarly, a marked decrease of all trace elements with the exception of copper occurs when rice is polished [176].

Large losses of nutrients also occur due to the canning of fruits and vegetables. Although the snap-freezing of vegetables is thought to preserve much of the vitamin content of such foods, it has been found, for example, that the freezing of vegetables results in a loss of 37–56% of vitamin B6 levels [177]. Levels of vitamin B6 and pantothenic acid are also decreased as a result of freezing and canning of fruits or fruit juices, with losses ranging from 7% to 50% [176]. Around 40 micronutrients including vitamins, essential minerals and trace elements are required in the human diet. Recommended dietary allowance of micronutrients is mostly based on information on acute effects of dietary deficiencies. However, for long-term health, the optimum intake of dietary micronutrient quantities is largely unknown, and a substantial percentage of the population is deficient in many of the micronutrients [177]. Many individual essential minerals have been found to be lacking in Western diets despite the availability of a huge range of foods, and supplementation of such minerals may prove to restore antioxidant balance in the body, or improve resistance

to AD via other mechanisms. For example, magnesium has been studied extensively by researchers and has been demonstrated to lower serum total cholesterol, decrease serum LDL and insulin-stimulated glucose uptake in type 1 diabetic subjects [178]. A more recent study of aged mice has found that a diet moderately deficient in magnesium, compared to a magnesium-supplemented diet, results in increased levels of oxidized lipids, and increased oxidative stress which was associated with inflammation [179].

## 16. Low-carbohydrate diets

In developing countries diets are increasingly becoming higher in trans-fats, refined foods and carbohydrates; however, levels of fibre have decreased. These dietary factors are contributing to a rapidly increasing prevalence of obesity and type 2 diabetes and a decline in health, particularly in the aged [149,180]. Alterations in dietary lipids have been posited as playing a role in cognitive defects in AD [146]. It may be likely that high-carbohydrate, high-trans-fat and high-cholesterol (HFHC) diets, together with declining levels of vitamin B12 and folate and with declines in trace minerals, play a role in the pathogenesis of AD [146].

The common advocacy of low-fat and high-carbohydrate diets is contradicted by various studies [150,151,181–184]. In fact, studies of glucose metabolism suggest this sort of diet is actually detrimental to human health. For example, in a dietary intervention study in which patients were subjected to either a high-fat or high-carbohydrate diet, it was demonstrated that triglyceride, glucose and insulin levels were higher on the high-carbohydrate diet [150]. In another dietary intervention study, it was shown that a high-carbohydrate diet led to increased insulin and triglycerides but to significantly lower levels of HDL [151]. Similar results were seen in non-insulin-dependent diabetes mellitus subjects when fed high- and low-carbohydrate diets; the conclusion suggested that high-carbohydrate diets did not improve glycemic control nor insulin sensitivity, but contributed to raised plasma triglycerides and VLDL concentrations yet reduced HDL levels [181]. Moreover, it has been demonstrated that a high-monounsaturated-fat, low-carbohydrate diet improves insulin sensitivity peripherally in non-insulin-dependent diabetes mellitus subjects [182].

The effect of high dietary fat on endurance performance in athletes has also been assessed, and a significantly improved performance was demonstrated when compared to a diet with high-carbohydrate intake [183]. Studies of aerobic exercise and diet on obese women have also demonstrated that a greater loss of weight occurs when subjects are on a low-carbohydrate diet than when consuming a low-fat diet [184]. The use of a low-fat, high-carbohydrate diet has also been found to accentuate hyperglycaemia and hyperinsulinaemia, therefore leading to upper-body obesity, glucose intolerance, hypertension and hypertriglyceridaemia (reviewed by Kaplan [185]). For example, a cross-sectional study of subjects in India revealed that central obesity was associated with higher postprandial plasma levels of insulin, glucose, serum iron and oxidative stress. This study also found that vitamin C, vitamin E, serum zinc/insulin ratio and serum magnesium/insulin ratio had an inverse association with high body fat [186].

Insulin resistance syndrome and associated conditions such as type 2 diabetes mellitus and hypertension are also associated with age-related memory impairment and AD

royalsocietypublishing.org/journal/rsob    Open Biol. **10**: 200084

[187,188]. As the evidence of links between insulin resistance, impaired glucose metabolism and AD increases, it would appear that diets appropriate for the prevention of diabetes and obesity may also be appropriate for the prevention of AD. A meta-analysis of a number of studies has also revealed that high-monounsaturated-fat diets can improve lipoprotein profiles and glycemic control in patients with type 2 diabetes [181]. Therefore, a diet rich in monounsaturated fatty acids and low in carbohydrates may be a useful addition for the elderly and also for AD patients.

In a study on the Mediterranean diet (MeDi) and risk for AD it was shown that adherence to the MeDi was associated with a lower risk of AD. This diet involves a high intake of vegetables, legumes, fruits, olive oil, fish and dairy, low meat, and moderate wine. [152]. High adherence has been attributed to centenarians in Sicily and they did not have any cardiac risk factors nor other major age-related diseases such as severe cognitive impairment or heart disease [189].

The effects of endogenous and exogenous antioxidants as derived from nutrition were comprehensively reviewed [155]. It has been shown in clinical research on humans that dietary interventions, including amino acids, vitamins, minerals and phytochemicals, have a substantial effect of glutathione levels, which thereby may provide clinical benefits [129]. Bruce N. Ames proposed that there are longevity vitamins and proteins that can be used as supplements for prolonging healthy ageing [15]. Moreover, seed sprouting improves the nutritional and antioxidant profiles and confers improved increases in protein, fats, vitamins, minerals and phyto-nutrients [190]. Furthermore, a reversing of memory problems in one hundred patients has recently been demonstrated in which a ketogenic diet among other interventions was used [19]. There is also a dietary guide book of nutrition for AD based on evidence from research [191].

# 17. Conclusion

Focusing research efforts, drug development strategies and healthcare approaches on a single component of a system, rather than the interacting network of components comprising such a system, may obscure important factors and/or disease mechanisms, including those evident during presymptomatic stages of disease. A holistic approach to healthcare to delay and prevent chronic diseases such as AD by lifestyle changes which optimize individual dietary needs is imperative. The MeDi has been shown to play a role in various factors involved in the pathogenesis of AD including oxidative stress and inflammation. Higher adherence to the diet by Sicilians has resulted in increased numbers of centenarians with a reduced risk for AD. Complex phenolic, carotenoid, antioxidants such as vitamin C and vitamin E are found in high concentrations in the MeDi [152,189].

In considering the multifactorial nature of AD, it may suggest that AD, like other neurodegenerative diseases and probably all degenerative diseases, may have a common link. Thus, the joint attack on the body by metabolic acidosis and oxidative stress may require a normalizing of extracellular and intracellular pH with simultaneous supplementation of a combination of antioxidants at sufficiently high personalized doses and a nutrient-rich, low-carbohydrate diet.

Data accessibility. This article has no additional data.

Authors' contributions. G.V. has written the paper. S.K.S. has collected the articles, corrected and edited the review, and arranged the references. G.P. performed proofreading and final editing.

Competing interests. Authors declare no conflict of interest.

Acknowledgement. This work was performed with resources from the non-government/non-profit organization Indian Scientific Education and Technology Foundation, Lucknow, India and Indian Council of Medical Research (ICMR), New Delhi, India.

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
