## [Reviewer comments · Open Biology]

Review History

RSOB-20-0084.R0 (Original submission)

Review form: Reviewer 1

Recommendation

Accept with minor revision (please list in comments)

Do you have any ethical concerns with this paper?

No

Comments to the Author

The authors in a manuscript entitled "Role of antioxidants and a nutrient-rich diet in Alzheimer's disease" have provided excellent information on how to prevent and maybe even reverse Alzheimer's disease.

Alzheimer's disease is summarised and Oxidative Stress is considered one of the major causes. Oxidative Stress and the modern diet both contribute to changes in the extracellular pH leading to chronic acidosis thereby resulting in a constant increase in free radical production and an increase in Amyloid plaques. Glutathione the major endogenous antioxidant is reviewed and how some nutrients can assist in increasing its levels. Astaxanthin the most potent carotenoid is expounded as well as Ascorbyl Palmitate a fat-soluble form of Vitamin C. Reference is made to

various animal studies which have shown the efficacy of these antioxidants. It suggests the use of a combination of antioxidants rather than a single antioxidant since they have a synergistic effect, as demonstrated in research. It also introduces the usage of Molecular Hydrogen, arguably the most potent and effective antioxidant in existence. Mention is also made of how agricultural methods have impacted on and caused lower nutrient levels in foods. Nutritional and nutraceutical usage is also addressed so that optimum dietary interventions can prevent the onset of Alzheimer's and possibly other chronic degenerative diseases. This is important as +/-70% of deaths globally are attributed to chronic diseases.

The article is informative and very well written and the Co-authors are highly respected with large numbers of publications in highly rated Scientific and Medical Journals. So, I recommend this review article for publication. The quality of the manuscript is suitable for publication. There are some minor issues which should be addressed:

- 1) Introduction should be more informative and elaborative.
- 2) Grammatical mistakes should be revised thoroughly.

Review form: Reviewer 2

Recommendation

Accept with minor revision (please list in comments)

Do you have any ethical concerns with this paper?

No

Comments to the Author

This review paper mainly discusses oxidative stress and metabolic acidosis in Alzheimer's disease (AD) and the effectiveness of antioxidants/diets/supplements. The topic is interesting and up-to-date.

A few suggestions are provided to improve the paper:

Authors discuss one neuropathological hallmark of AD, amyloid beta deposits and its relation with oxidative stress. But the disease is also characterized by tau pathology, which should be included in the paper. Also, a brief discussion about the relation between tau pathology and oxidative stress/damage and pH alterations can be included in the paper.

Authors state that "... treatment in degenerative diseases, including AD, may require a normalizing of extracellular and intracellular pH with a simultaneous supplementation of an antioxidant combination cocktail at a sufficiently high dose". It is known that high-dose antioxidant supplements may be harmful in some cases. How can we find the equilibrium in chronic diseases like AD?

The authors discuss several observations made in their own laboratory. The inclusion of figures to support the discussion of the authors' findings would benefit the paper.

A brief discussion about the mediterranean diet and centenarians/lower risk for chronic diseases (namely AD) could be included in the paper.

Earthing or Grounding paragraph can be removed.

In the conclusions, the authors should include their opinion about the preventive as well as therapeutic effects of the several diets or supplements discussed in the paper, particularly in the context of brain aging and dementia (AD).

There are some typos and language errors throughout the text.

Review form: Reviewer 3

Recommendation

Major revision is needed (please make suggestions in comments)

Do you have any ethical concerns with this paper?

No

Comments to the Author

The review entitled "Role of antioxidants and a nutrient rich diet in Alzheimer's disease" is devoted to a quite important and actual topic of prevention and neuroprotection at the most common neurodegenerative disorder. However, the paper has some major limitations as below:

Major points:

1) The manuscript is poorly structured. There is no clear link between its parts and some of the parts logically overlap. The structure should be carefully revised. Authors should highlight major themes and support them with more narrow and specified topics. The conclusions and hypotheses should be evidence-based. Logical transitions between the parts should be presented.

2) Although some parts are well written and informative (for instance, Molecular Hydrogen), most of the text is confusing. Major part of the manuscript represents the notions and results of the 1990s or even earlier and so they are out of date. Since this is not a historical report, authors should refer to current findings.

3) There are statements and conclusions that are not supported by the controlled experimental or clinical findings. For instance, in the part entitled "Effect of Diet on extracellular pH" they state "humans develop a progressive increase in extracellular acidity" and "by eating more fruits and non-grain plant foods, may hold benefits for preventing or delaying many of the diet and age-related degenerative diseases and their consequences". However, there is no any prove of metabolic acidosis in Alzheimer's disease or in elderly population, nor of the effects of fruit-rich diet on the pH in vivo. There are certain pathological conditions such as diabetic ketoacidosis or brain ischemia that do produce metabolic acidosis in the brain tissue. Moreover, diabetic ketoacidosis increases risk of Alzheimer's disease. Therefore, the relation between pH and Alzheimer's disease is not that simple and direct as it was considered in the 1990s. The authors should refer to the recent findings in the field and revise this part in accordance.

Similarly, the evaluation of the effects of exogenous antioxidants in Alzheimer's disease has changed significantly since the 1990s. Current systematic reviews and meta-analyses showed no convincing evidence of vitamin and mineral supplementation on clinical manifestations or neuropathology of Alzheimer's disease, nor for preventing dementia or delaying cognitive decline in people with mild cognitive impairment. The focus now is biased to the regulation and modulation of the inner antioxidant system, namely, Keap1-Nrf2 axis. The authors should refer to the current research in the field and up-to-date notions on the pathophysiological mechanisms in Alzheimer's disease, including oxidative glutamate toxicity.

4) It does not seem appropriate to include the part entitled "Grounding" into the review on antioxidants and a nutrient rich diet since the mechanisms of this alternative medicine approach

on human organism are rather questionable. There is no any experimental proof for the mechanism mentioned in the review (a flow of electrons from Earth to human body). On the contrary, accurate analysis of the charge exchange between the human body and ground showed that currents between the environment (earth) and a grounded human body are very small (nanoamperes) and do not appear to contain information other than information about subject motion (please, refer to Chamberlin et al., J Chiropr Med., 2014; doi: 10.1016/j.jcm.2014.10.001).

5) The parts entitled "Agricultural methods" and "Food processing and storage methods" appear to be slightly related to the review topic. In addition, they review the data of the 1990s and early 2000s.

6) There are three inclusions of the authors' unpublished results into the text. Since there are three co-authors of the manuscript with different affiliations, it would be desirable to specify the author rather than to write "In my unpublished research...". Usually such inclusions illustrate and support some of the authors' hypotheses or statements. For this purpose, some of the particular experimental results relevant to the topic are presented as figures or tables with statistically significant differences and a brief description of the methods applied. The presentation of the authors' unpublished data as a short mention does not seem sufficient (P. 15 and P. 24).

Minor points:

1) P. 6. Kg Dolton should be corrected to kiloDalton or kDa.

2) Reference list is issued carelessly. It should be thoroughly revised and unified according to the Journal's requirements.

Decision letter (RSOB-20-0084.R0)

01-May-2020

Dear Dr Singh,

We are pleased to inform you that your manuscript RSOB-20-0084 entitled "Role of antioxidants and a nutrient rich diet in Alzheimer's disease" has been accepted by the Editor for publication in Open Biology. The reviewer(s) have recommended publication, but also suggest some minor revisions to your manuscript. Therefore, we invite you to respond to the reviewer(s)' comments and revise your manuscript.

Please submit the revised version of your manuscript within 14 days. If you do not think you will be able to meet this date please let us know immediately and we can extend this deadline for you.

- 1) A text file of the manuscript (doc, txt, rtf or tex), including the references, tables (including captions) and figure captions. Please remove any tracked changes from the text before submission. PDF files are not an accepted format for the "Main Document".
- 2) A separate electronic file of each figure (tiff, EPS or print-quality PDF preferred). The format should be produced directly from original creation package, or original software format. Please note that PowerPoint files are not accepted.
- 3) Electronic supplementary material: this should be contained in a separate file from the main text and meet our ESM criteria (see <http://royalsocietypublishing.org/instructions-authors#question5>). All supplementary materials accompanying an accepted article will be treated as in their final form. They will be published alongside the paper on the journal website and posted on the online figshare repository. Files on figshare will be made available approximately one week before the accompanying article so that the supplementary material can be attributed a unique DOI.

Online supplementary material will also carry the title and description provided during submission, so please ensure these are accurate and informative. Note that the Royal Society will not edit or typeset supplementary material and it will be hosted as provided. Please ensure that the supplementary material includes the paper details (authors, title, journal name, article DOI). Your article DOI will be 10.1098/rsob.2016[*last 4 digits of e.g. 10.1098/rsob.20160049*].

- 4) A media summary: a short non-technical summary (up to 100 words) of the key findings/importance of your manuscript. Please try to write in simple English, avoid jargon, explain the importance of the topic, outline the main implications and describe why this topic is newsworthy.

Images

Data-Sharing

It is a condition of publication that data supporting your paper are made available. Data should be made available either in the electronic supplementary material or through an appropriate repository. Details of how to access data should be included in your paper. Please see <http://royalsocietypublishing.org/site/authors/policy.xhtml#question6> for more details.

Data accessibility section

Sincerely,

The Open Biology Team
mailto:openbiology@royalsociety.org

Reviewer(s)' Comments to Author:

Referee: 1

Comments to the Author(s)

The authors in a manuscript entitled "Role of antioxidants and a nutrient-rich diet in Alzheimer's disease" have provided excellent information on how to prevent and maybe even reverse Alzheimer's disease.

Alzheimer's disease is summarised and Oxidative Stress is considered one of the major causes. Oxidative Stress and the modern diet both contribute to changes in the extracellular pH leading to chronic acidosis thereby resulting in a constant increase in free radical production and an increase in Amyloid plaques. Glutathione the major endogenous antioxidant is reviewed and how some nutrients can assist in increasing its levels. Astaxanthin the most potent carotenoid is expounded as well as Ascorbyl Palmitate a fat-soluble form of Vitamin C. Reference is made to various animal studies which have shown the efficacy of these antioxidants. It suggests the use of a combination of antioxidants rather than a single antioxidant since they have a synergistic effect, as demonstrated in research. It also introduces the usage of Molecular Hydrogen, arguably the most potent and effective antioxidant in existence. Mention is also made of how agricultural methods have impacted on and caused lower nutrient levels in foods. Nutritional and nutraceutical usage is also addressed so that optimum dietary interventions can prevent the onset of Alzheimer's and possibly other chronic degenerative diseases. This is important as +/-70% of deaths globally are attributed to chronic diseases.

The article is informative and very well written and the Co-authors are highly respected with large numbers of publications in highly rated Scientific and Medical Journals. So, I recommend this review article for publication. The quality of the manuscript is suitable for publication. There are some minor issues which should be addressed:

- 1) Introduction should be more informative and elaborative.
- 2) Grammatical mistakes should be revised thoroughly.

Referee: 2

Comments to the Author(s)

This review paper mainly discusses oxidative stress and metabolic acidosis in Alzheimer's disease (AD) and the effectiveness of antioxidants/diets/supplements. The topic is interesting and up-to-date.

A few suggestions are provided to improve the paper:

Authors discuss one neuropathological hallmark of AD, amyloid beta deposits and its relation with oxidative stress. But the disease is also characterized by tau pathology, which should be included in the paper. Also, a brief discussion about the relation between tau pathology and oxidative stress/damage and pH alterations can be included in the paper.

Authors state that “... treatment in degenerative diseases, including AD, may require a normalizing of extracellular and intracellular pH with a simultaneous supplementation of an antioxidant combination cocktail at a sufficiently high dose”. It is known that high-dose antioxidant supplements may be harmful in some cases. How can we find the equilibrium in chronic diseases like AD?

The authors discuss several observations made in their own laboratory. The inclusion of figures to support the discussion of the authors’ findings would benefit the paper.

A brief discussion about the mediterranean diet and centenarians/lower risk for chronic diseases (namely AD) could be included in the paper.

Earthing or Grounding paragraph can be removed.

In the conclusions, the authors should include their opinion about the preventive as well as therapeutic effects of the several diets or supplements discussed in the paper, particularly in the context of brain aging and dementia (AD).

There are some typos and language errors throughout the text.

Referee 3:

The review entitled “Role of antioxidants and a nutrient rich diet in Alzheimer’s disease” is devoted to a quite important and actual topic of prevention and neuroprotection at the most common neurodegenerative disorder. However, there are some issues that could be addressed:

Major points:

1) The structure could benefit from some careful revision. There is no clear link between its parts and some of the parts logically overlap. Authors should highlight major themes and support them with more narrow and specified topics. The conclusions and hypotheses should be evidence-based. Logical transitions between the parts should be presented.

2) There are statements and conclusions that are not supported by the controlled experimental or clinical findings. For instance, in the part entitled “Effect of Diet on extracellular pH” they state “humans develop a progressive increase in extracellular acidity” and “by eating more fruits and non-grain plant foods, may hold benefits for preventing or delaying many of the diet and age-related degenerative diseases and their consequences”. However, there is no any prove of metabolic acidosis in Alzheimer’s disease or in elderly population, nor of the effects of fruit-rich diet on the pH in vivo. There are certain pathological conditions such as diabetic ketoacidosis or brain ischemia that do produce metabolic acidosis in the brain tissue. Moreover, diabetic ketoacidosis increases risk of Alzheimer’s disease. Therefore, the relation between pH and Alzheimer’s disease is not that simple and direct as it was considered in the 1990s. The authors should refer to the recent findings in the field and revise this part in accordance.

Similarly, the evaluation of the effects of exogenous antioxidants in Alzheimer’s disease has changed significantly since the 1990s. Current systematic reviews and meta-analyses showed no convincing evidence of vitamin and mineral supplementation on clinical manifestations or neuropathology of Alzheimer’s disease, nor for preventing dementia or delaying cognitive decline in people with mild cognitive impairment. The focus now is biased to the regulation and modulation of the inner antioxidant system, namely, Keap1-Nrf2 axis. The authors should refer to the current research in the field and up-to-date notions on the pathophysiological mechanisms in Alzheimer’s disease, including oxidative glutamate toxicity.

3) It does not seem appropriate to include the part entitled “Grounding” into the review on antioxidants and a nutrient rich diet since the mechanisms of this alternative medicine approach

on human organism are rather questionable. There is no any experimental proof for the mechanism mentioned in the review (a flow of electrons from Earth to human body). On the contrary, accurate analysis of the charge exchange between the human body and ground showed that currents between the environment (earth) and a grounded human body are very small (nanoamperes) and do not appear to contain information other than information about subject motion (please, refer to Chamberlin et al., J Chiropr Med., 2014; doi: 10.1016/j.jcm.2014.10.001).

4) The parts entitled "Agricultural methods" and "Food processing and storage methods" appear to be slightly related to the review topic. In addition, they review the data of the 1990s and early 2000s.

5) There are three inclusions of the authors' unpublished results into the text. Since there are three co-authors of the manuscript with different affiliations, it would be desirable to specify the author rather than to write "In my unpublished research...". Usually such inclusions illustrate and support some of the authors' hypotheses or statements. For this purpose, some of the particular experimental results relevant to the topic are presented as figures or tables with statistically significant differences and a brief description of the methods applied. The presentation of the authors' unpublished data as a short mention does not seem sufficient (P. 15 and P. 24).

Minor point: P. 6. Kg Dolton should be corrected to kiloDalton or kDa.

Decision letter (RSOB-20-0084.R1)

19-May-2020

Dear Dr Singh

We are pleased to inform you that your manuscript entitled "Role of antioxidants and a nutrient rich diet in Alzheimer's disease" has been accepted by the Editor for publication in Open Biology.

Sincerely,
The Open Biology Team
mailto: openbiology@royalsociety.org